# Effect of the Surface Morphology of Porous Coatings on Secondary Electron Yield of Metal Surface

**DOI:** 10.3390/ma15124322

**Published:** 2022-06-18

**Authors:** Min Peng, Shu Lin, Chuxian Zhang, Haifeng Liang, Chunliang Liu, Meng Cao, Wenbo Hu, Yonggui Zhai, Yongdong Li

**Affiliations:** 1Key Laboratory for Physical Electronics and Devices of the Ministry of Education, School of Electronic Science and Engineering, Xi’an Jiaotong University, Xi’an 710049, China; l623121wxn@stu.xjtu.edu.cn (M.P.); shulin@mail.xjtu.edu.cn (S.L.); chlliu@mail.xjtu.edu.cn (C.L.); mengcao@mail.xjtu.edu.cn (M.C.); huwb@mail.xjtu.edu.cn (W.H.); zhaiyg@mail.xjtu.edu.cn (Y.Z.); 2Beijing Institute of Aerospace Micro-Electromechanical Technology, China Research Institute of Aerospace Electronic Technology, Beijing 100094, China; chuxian@126.com; 3Key Laboratory for Optical Measurement and Thin Films of Shaanxi Province, Xi’an Technological University, Xi’an 710032, China; lianghaifeng@xatu.edu.cn

**Keywords:** secondary electron yield (SEY), surface morphology, porous coatings, wet etching

## Abstract

Surface roughening is an important material surface treatment technique, and it is particularly useful for use in secondary electron yield (SEY) suppression on metal surfaces. Porous structures produced via roughening on coatings have been confirmed to reduce SEY, but the regulation strategy and the influence of process parameters both remain unclear in the practical fabrication of effective porous structures. In this paper, the effect of the surface morphology of porous coatings on the SEY of aluminum alloy substrates was studied. Surface characterization and SEY measurements were carried out for samples with a specific process technique on their surfaces. An exponential fitting model of the correlation between surface roughness and the peak values of SEY curves, $\delta_{m}$, was summarized. Furthermore, an implementation strategy to enable low surface SEY was achieved from the analysis of the effect of process parameters on surface morphology formation. This work will aid our understanding of the effect of the irregular surface morphology of porous coatings on SEY, thereby revealing low-cost access to the realization of an easy-to-scale process that enables low SEY.

## 1. Introduction

Secondary electron yield (SEY) is a yardstick used to weigh the magnitude of secondary electron emission (SEE) due to incident electron bombardment, and high SEY may lead to the multipactor effect, which causes serious problems such as reduced reliability, power loss and even the failure of aerospace microwave devices [1,2,3,4,5]. This undesired effect is a deleterious electron avalanche [6,7] that can be suppressed by the surface treatment, which reduces SEY in practical engineering. This necessitates a fundamental understanding of the various factors that affect SEY and the approaches used to change these factors via surface treatment. In this work, understanding the effect of surface morphology on surface SEY by changing parameters in the treatment process is of key importance.

Surface treatment on metals has long been researched, mainly including Alodine coatings [8,9], inert metal coatings, surface amorphous carbonation [10,11,12,13,14,15,16,17] and roughening [18,19,20,21,22,23,24,25]. In the field of rough surfaces, some aspects, such as calculations and mechanisms, have been well-defined [26,27] while experimental studies continue to prosper. Montero et al. [16] introduced the roughness effect when studying the SEE properties of graphene nanoplatelets. Surface roughness evolution, which induced low SEY in carbon-coated Ag/Al substrates, was also investigated [28]. In recent years, Nistor V. et al. [29] were the first to propose a microstructured gold/silver coatings scheme, and it was shown that the obtained stable porous structure could inhibit the SEY, reducing it to about 1.0, in which the thickness of the gold plates was 2 µm. This was the first design of a method that used a combination of metal coating and roughening; however, the appearance of the gold/silver alloy was later thought to be harmful to device performance by D.D. Wu et al. modified the microstructured gold/silver coatings scheme by adding TiO_2_ between the gold and silver [30]. Furthermore, Montero I. et al. [31] achieved low SEY by applying rough silver coatings to RF filters prone to multipactor discharge. In the studies discussed above, all of the roughness was provided by the porosity of etched Ag. The porous silver structure scheme was considered to be the most promising process, but previous studies have not revealed a substantive basis for the implementation of parameters or a regulative strategy for low SEY. Experimental reports regarding the effect of different schemes of the process parameters on SEY for porous gold/silver coatings are still scarce.

Therefore, the research presented here takes this further. The goal of this work was to explore the cascading effects of process parameters, the surface morphology of the resultant porous coatings and their surface SEY in turn. More specifically, we were interested in determining how different process parameters affected the etching degree to make the characteristics of surface morphology differ and how these morphology characteristics affected the surface SEY. Therefore, in this work, we used a combined process scheme of coating and wet etching to prepare the porous coating. We characterized the surface morphology of the porous coating on samples under different process parameter schemes and measured the corresponding SEY curves as the function of primary electron energy to deduce what type of surface morphology can suppress SEY and what parameter schemes produce this morphology. In this work, we provided an insight into the effect of process parameters and the surface morphologies of porous coatings on SEY. This provided abundant data, as well as a complete description of surface roughness on porous coatings, which aid in gaining a low-cost process strategy to achieve low SEY.

## 2. Materials and Methods

In this experimental preparation, the aluminum alloy 6061, which is one of the most widely used aluminum alloys in aerospace devices, was chosen as a substrate for the samples. It was machined into pieces with diameters of 20 mm × 15 mm × 1 mm, and a three-step process of plating (Ni/Ag)–wet chemical etching–nanoscale gold coating was then performed to obtain porous coating on the surfaces of these samples. As shown in Figure 1, firstly, a 10 μm nickel layer and a 30 μm silver layer were coated via electroplating, in which the silver layer was etched to achieve high aspect ratio surface roughness and electrical conductivity, while the nickel layer was added between the silver layer and the substrate to increase the adhesion. After that, these prepared samples were wet etched in a mixed acid solution of diluted nitric acid (65 wt% HNO_3_), hydrofluoric acid (48 wt% HF) and deionized water (18 MΩ) with the composition ratio of 1:0.288:2.712 in a high-density polyethylene container. In this step, the etching temperature was changed from 15 °C to 40 °C using a water bath, and the etching lasted from 90 to 300 s. The control variable method was used when changing the parameters of the etching conditions, which were etching temperature and duration, to study the effects of different surface morphologies on SEY because the etching intensity manipulated by these parameters affected the surface morphology. All of the parameter schemes for the etching process were labeled using the tags presented in Table 1. After etching was complete, the samples were placed in an ultrasonic bath and underwent nitrogen drying, and at the end of the ultrasonic bath, the disappearance of the white resultant attached to the surface was observed. Finally, gold coatings with different thicknesses of 100 nm and 500 nm were deposited within the ±50 nm error range via magnetron sputtering. It was decided that the gold coatings would have nanoscale thicknesses to balance the benefits and risks of gold/silver alloy in order to prevent the porous silver surface from being oxidized. The DC power was set to 50 W, and the working pressure was set to 0.45 Pa. In Figure 1, the cross-sectional structure of the final multilayer samples is presented in an enlarged view.

Moreover, in this work, the surface morphology was observed using a scanning electron microscope (SEM) (Gemini SEM 500, ZEISS, Jena, Germany), and the surface roughness was measured using a three-dimensional color laser scanning microscope (VK9700K, KEYENCE, Osaka, Japan). The secondary electron yield measurement was carried out using a homemade SEY system, which can be adapted for the SEY measurement of insulators [32].

## 3. Results

### 3.1. The Characterization of Surface Morphology

The observation of samples’ surfaces from SEM images provides microscopic details of porous coatings with which the characteristics of surface morphology can be analyzed. Figure 2 presents the evolution of samples’ surface morphologies in the process using the parameter scheme *D4*. The surface of the sample with nickel and silver layers in Figure 2a appeared to be plate-like and smooth before wet etching. In Figure 2b, the microstructures produced via etching made the surface rugged and porous. After gold sputtering, the edges and corners of the porous microstructures become gentle and rounded into sphere-like, granular clusters, as shown in Figure 2c. However, this characteristic of granular morphology was not universal in all of the schemes. Figure 3 shows the porous microstructures before and after gold coating was obtained using different schemes, and these schemes led to a tremendous difference in surface morphology. The microstructure produced using the *A3* scheme before gold coating had islands and a flat top, while the structures produced using the *D4* scheme appeared to have porous granular morphologies with uniformity. The resulting porous structures from different schemes were different in terms of porosity and particle size, which was reflected in the difference in morphology before gold coating between the *E2* scheme and the *D5* scheme. After gold coating, wrapping and filling with the gold layer made the characteristics of surface morphology more obvious. However, the application of a 500 nm gold layer on the porous structures of the *D5* scheme made it become granular, and fine structures disappeared. With a further increase in the thickness of the gold layer, the discrepancy in the surface morphology due to this thickness could potentially affect the surface SEY that corresponded to different schemes.

In order to quantitatively describe the difference between these morphological features, a parameter $p_{c}$, called the characteristic pore size of a porous structure, was defined as the average distance between adjacent visible particles in a sampling area on porous surface structures. It could be calculated using Equation (1) according to SEM images, where *n* is the number of pores and $p_{i}$ represents the size of each individual pore which is usually on the micron scale. *i* circulates from 1 to *n* to calculate the average size of n pores in an image, and the result is accurate to one decimal place. Thus, the $p_{c}$ values for the SEY images corresponding to different schemes could be calculated, as shown in Table 2.
(1)$$p_{c}=\frac{1}{n}\overset{n}{\underset{i=1}{\sum }}p_{i}$$

While $p_{c}$ reflects the real appearance of porous structures, it cannot be obtained using large-scale measurements. Therefore, the values of the absolute average relative to the base height of 3D surface profiles, $R_{a}$, were used to characterize surface roughness. The value of $R_{a}$ was obtained by extracting surface height information measured using a color laser scanning microscope (LSM), and it was calculated according to Formula (2). $R_{a}$ provided the average difference between peaks and valleys. It was more appropriate for use in the macroscopic evaluation of surface morphology, while $p_{c}$ was advantageous to use in the analysis of local microporous structures. An example of gaining $R_{a}$ from a surface profile is shown in Figure 4, which outlines the process of extracting and processing profile information. This underpins the average $R_{a}$ values of some representative schemes presented in Table 2. It should be noted that $R_{a}$ reflected the arithmetic average of the absolute values of the deviation from the points on the actual profile to its profile centerline within a selected area. Additionally, the $R_{a}$ values for schemes in Table 2 were averages of the $R_{a}$ measurements of several substrates with this scheme. The $R_{a}$ measurement of one substrate was also an average of five sampling areas on a same substrate, which eliminated the effect of a random defect in one sampling area.
(2)$$R_{a}=\frac{1}{n}\overset{n}{\underset{i=1}{\sum }}\left|y_{i}-\bar{y}\right|$$

### 3.2. The Measurements of Secondary Electron Yield

The SEY measurement system is outlined in Figure 5. The electron beam irradiated the target sample surface with primary energy from 30 eV to 3000 eV, which could be adjusted by the output of a DC power. The emitted SEs were entirely collected by the collector biased with a positive potential of 40 V. The secondary electrons current $I_{s}$ and the target current $I_{t}$ could be measured using the continuously irradiated method, and then the primary electron current $I_{p}$ was obtained as $I_{p}=I_{s}+I_{t}$. The SEY value *δ* could finally be calculated using:(3)$$\delta =\frac{I_{s}}{I_{p}}=\frac{I_{s}}{I_{s}+I_{t}}$$

Figure 6 presents the SEY results of substrates before and after the surface treatment with some different process schemes. Notably, the SEY result for each scheme is an average of the SEY measurements of five samples, and the measured value of each sample is also an average of the results of several sampling areas selected on it. It was found that the $\delta_{m}$, the average value of the maximum SEY as a function of the primary electron energy, decreased a lot after the treatment. The $\delta_{m}$ for substrates decreased from 2.2 (with the error of 0.4) to 1.1 (within ±0.03) ~1.2 with the *D4* scheme for the 100 nm gold batch (*D4-100n*) and the *D4* scheme for the 500 nm gold one (*D4-500n*), in which the best SEY suppression reached 54% in the *D4-100n* scheme. For more evidence to indicate the great differences in the final $\delta_{m}$ values caused by process schemes, the $\delta_{m}$ results corresponding to ten schemes with 100 nm-thick gold coatings are shown in Table 3. More results of $\delta_{m}$ for schemes in Table 1 are shown in Appendix A
Table A1 and Table A2.

## 4. Discussion

In this paper, a preliminary exploration of the effect of surface morphology on the SEY of metal surfaces was carried out. Based on an analysis of surface characterization and SEY measurements, the common surface characteristics for low SEY were found to be granular clusters with $p_{c}$ values around 1.0 μm as well as large surface $R_{a}$ values (at least above 0.6 μm). Furthermore, we obtained a model for the relationship between *δ_m_* and $R_{a}$ by fitting the values for samples with 100 nm gold coatings, as shown in Figure 7. The model was given by Equation (4) with $\delta_{0}=1.45\pm 0.07$, *b* = 0.18 ± 0.20${\;\mu m}^{-1}$ and *c* = −0.37 ± 0.16. The adjusted *R*^2^ for this fitting model was 0.9954, indicating a good description of the trend of this correlation. For robust validation, more $R_{a}$ data from additional experiments were put into this fitting model, and the corresponding predicted $\delta_{m}$ values that were obtained are presented in Table 4. The deviations were within 5% when compared with measured values. Moreover, the statistical $R_{a}$ data for different schemes are highlighted by colorful blocks, and we clearly notice that the $\delta_{m}$ value decreased dramatically with the increase in $R_{a}$. This agrees with the well-known SE capturing effect via microstructures.
(4)$$\delta_{m}\left(R_{a}\right)=\delta_{0}\exp\left(bR_{a}+cR_{a}^{2}\right)$$

This fitting model allows for further extrapolations to be made and can be used to predict the subtle correlation between the roughness and SEY of porous coatings on metal substrates, and it also helps one to understand what kind of surface morphology can suppress the SEY in this technological process.

Furthermore, from the perspective of practical engineering, we paid more attention to the process schemes that can produce morphologies conducive to suppressing SEY. From Figure 7, it can be seen that the *D4* scheme seemed to produce the best embodiment of this morphology. The impacts of the etching parameters—duration and temperature—on $\delta_{m}$ are presented in Figure 8 before and after 100 nm gold coating. From this figure, we can see that more potential schemes can be provided to achieve low SEY when etching temperatures are 25~35 °C. Additionally, it shows that a slight reaction cannot produce a morphology with low SEY when the values of the etching parameters are extremely small. The 100 nm gold coating was shown to narrow the distribution range of $\delta_{m}$ caused by the disparity of morphologies through a comparison before and after gold coating.

The effect of the parameters in the process scheme on the surface morphology lay in the intensity and degree of wet etching as well as the filling degree for pores via gold sputtering, which was a synthetical outcome of the morphology regulation by all parameters. A variation in every single parameter had a limited impact on SEY. Figure 9 presents some correspondence between the etching parameter schemes, the $R_{a}$ values of resultant morphologies and corresponding SEY suppression percentages for a 100 nm-thick gold coating. From this figure, we can see that when $R_{a}$ has a large value in the upper plane, the corresponding percentage of SEY suppression in the lower plane is also high. This figure directly displays the correlation between $R_{a}\;$ and $\delta_{m}$. More significantly, it provides a basis for further research into parameter schemes that can produce surface morphologies that enable low SEY. Our attention can be focused on the parameter region corresponding to large $R_{a}$ values. This region is roughly concentrated in the center due to the need for a proper etching temperature and duration to attain a large $R_{a}$ to suppress SEY. A low etching temperature and a short etching duration could make the etching degree too weak to reach an appropriate $R_{a}$ while a temperature that is too high and a duration that is too long leads to excessive etching and the disappearance of the rough structures that were formed. More schemes within this region that were not discussed in this paper remain to be explored.

## 5. Conclusions

In summary, in this paper, we began by presenting an experimental study of the cascading correlation between process parameters, surface morphology and SEY, and then the effect of the surface morphology of porous coatings on the SEY of metal surfaces was analyzed. The optimal process parameters of porous coatings for SEY suppression were summarized for applications in high-power, space-borne devices. This contributes to the creation of a process strategy that could be used to achieve low SEY.

In addition, an empirical model of the relationship between the average deviation value, $R_{a}$, of the surface profile and surface SEY was created, which can be used as a baseline for the prediction of low SEY. This model can be used to guide surface treatment strategies to obtain an optimal surface morphology that enables lower SEY.

## Figures and Tables

**Figure 1 materials-15-04322-f001:**
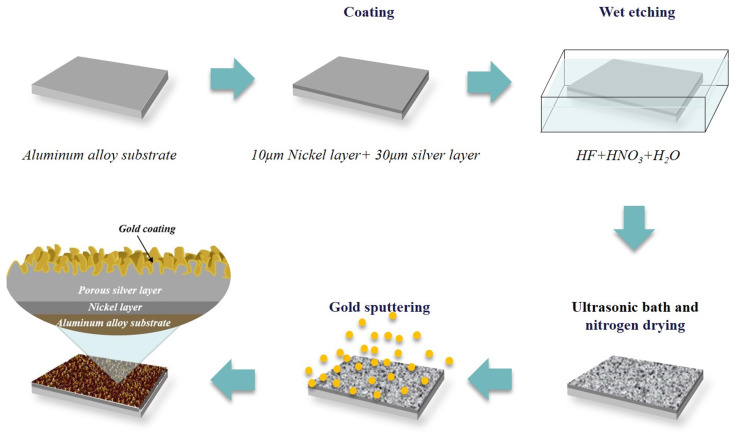
Schematic diagram of the sample structural evolution throughout the whole process.

**Figure 2 materials-15-04322-f002:**
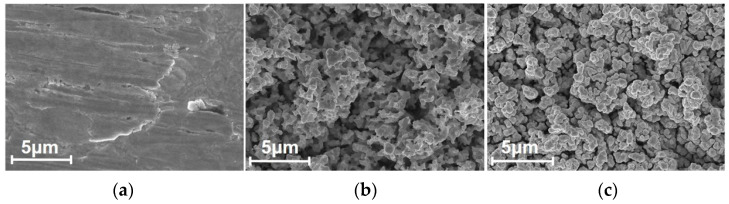
The SEM images of the *D4* scheme sample surface throughout experimental process. (**a**) The surface morphology before wet etching. (**b**) The surface morphology after wet etching. (**c**) The surface morphology after gold coating.

**Figure 3 materials-15-04322-f003:**
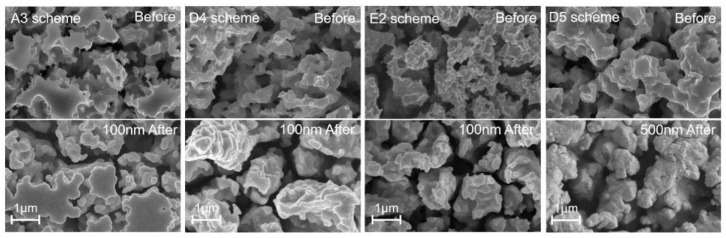
The SEM images of porous structures on samples using different schemes before and after the 100 nm/500 nm gold coating.

**Figure 4 materials-15-04322-f004:**
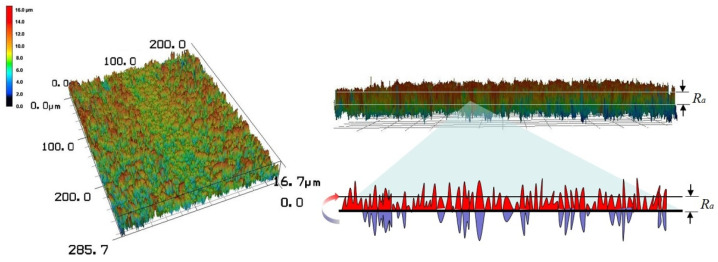
An example of three-dimensional surface profile and the corresponding $R_{a}$ for one sampling area on a substrate with *D4-100n* scheme.

**Figure 5 materials-15-04322-f005:**
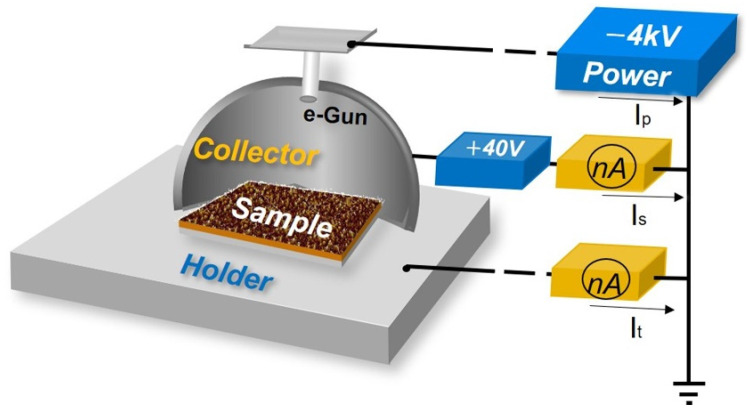
The sketch of SEY measurement system.

**Figure 6 materials-15-04322-f006:**
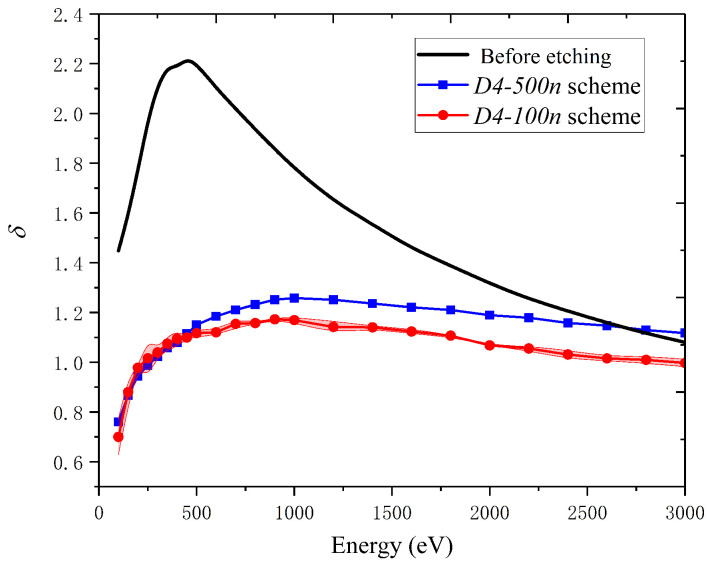
Measured SEY curves as a function of primary electron energy before and after the etching and gold coating. The red shadow area with dotted line in it represents the error from multiple measurements.

**Figure 7 materials-15-04322-f007:**
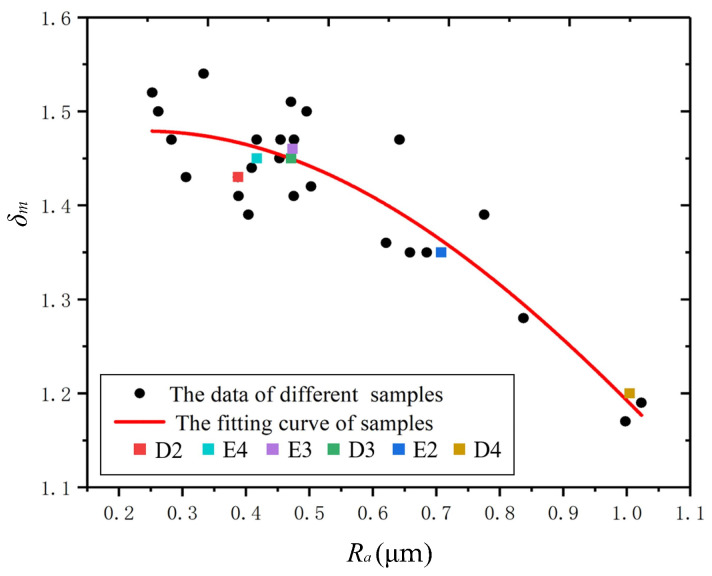
The correlation between $\delta_{m}$ and $R_{a}$ values for 100 nm gold samples under different schemes.

**Figure 8 materials-15-04322-f008:**
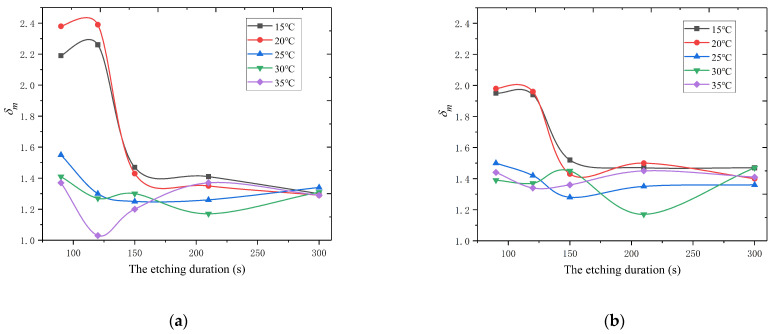
The dependence of surface $\delta_{m}$ values on etching parameters. (**a**) Before gold coating; (**b**) after 100 nm gold coating.

**Figure 9 materials-15-04322-f009:**
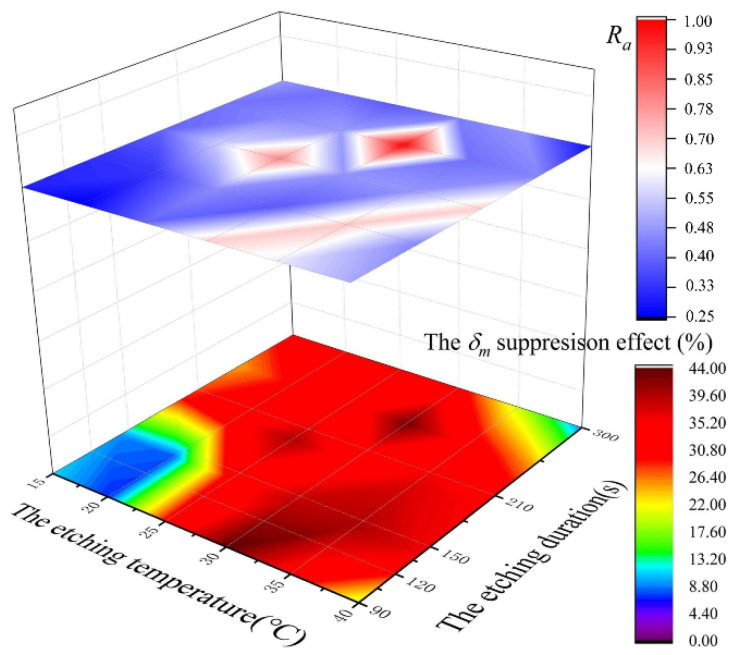
The relations among parameter schemes, corresponding morphologies and final SEY suppression effect.

**Table 1 materials-15-04322-t001:** The labels for the schemes of experimental process parameters.

Parameters	15 °C	20 °C	25 °C	30 °C	35 °C	40 °C
90 s	*A* *1*	*B* *1*	*C1*	*D1*	*E1*	*F1*
120 s	*A2*	*B* *2*	*C2*	*D2*	*E2*	*F2*
150 s	*A3*	*B* *3*	*C3*	*D3*	*E3*	*F3*
210 s	*A4*	*B4*	*C4*	*D4*	*E4*	*F4*
300 s	*A5*	*B5*	*C5*	*D5*	*E5*	*F5*

**Table 2 materials-15-04322-t002:** The *p_c_* and average $R_{a}$ values of the samples treated with different parameter schemes.

Schemes Label	*D2-100n* *	*E2-100n*	*D3-100n*	*E3-100n*	*D4-100n*	*E4-100n*
$P_{c}$ (μm)	0.9	1.3	1.2	1.2	1.0	1.1
$R_{a}$ (μm)	0.3876	0.7075	0.4712	0.4733	1.0046	0.4170

* *-100n* represents 100 nm-thick gold coating.

**Table 3 materials-15-04322-t003:** The *δ_m_* values of the substrates treated with different schemes of process parameters.

Scheme	$\delta_{m}$	Scheme	$\delta_{m}$	Scheme	$\delta_{m}$	Scheme	$\delta_{m}$
*D2-100n* *	1.39	*E2-100n*	1.34	*D2-500n* *	2.10	*E2-500n*	1.44
*D3-100n*	1.40	*E3-100n*	1.36	*D3-500n*	1.95	*E3-500n*	1.48
*D4-100n*	1.17	*E4-100n*	1.35	*D4-500n*	1.26	*E4-500n*	1.56
*D5-100n*	1.47	*E5-100n*	1.41	*D5-500n*	1.50	*E5-500n*	2.03

* *-100n* and *-500n* represent 100 nm- and 500 nm-thick gold coatings, respectively.

**Table 4 materials-15-04322-t004:** The additional experimental data of $R_{a}$ and their corresponding SEY values.

*$R_{a}$* (μm)	Predicted SEY Values	Measured SEY Values	The Relative Deviation
0.97	1.21	1.22	0.82%
1.06	1.15	1.20	4.17%
1.15	1.08	1.13	4.42%

## Data Availability

The data used to generate the curves shown are available on request.

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
