# Peer review of "Effect of the Surface Morphology of Porous Coatings on Secondary Electron Yield of Metal Surface"

_materials, 2022, doi:10.3390/ma15124322_

Round 1
Reviewer 1 Report
This manuscript is an investigation of the relationship between surface roughening/morphology on aluminium alloy substrates and secondary electron yield suppression (SEY). It is quite well written, particularly the first half of the manuscript, and very interesting. Figures are generally well done.
The Introduction is concise and effectively introduces both the project and aims, and the relevant literature. The authors state that “… this is the first work we know of that gives an insight into the surface roughness of porous coatings …” Without reading either the literature or the remainder of the manuscript, the relevance of porosity is unclear. Perhaps this could be clarified in the Introduction. Possible typo: line 47, “The Porous …” to “The porous …”
The Materials and Methods provides good coverage of the experimental and Figure 1 is an excellent diagram of the process.
In Table 2., in The Characterization of Surface Morphology section, data for the pore size parameters, pc and surface roughness, Ra are given for a subset of the parameter schemes. It is not clear how Ra is calculated (shown in Figure 4) and the meaning of the magnitudes of these values. This should be clarified and a full listing of these values for all schemes would be valuable. Also, the symbol Ra should be in italics.
Nice overview in The Measurements of Secondary Electron Yield section. Table 3. lists a selection of the SEY maximum values – again it would be helpful to have a full listing, perhaps in a Supplementary Information file.
The Discussion describes interesting results and correlations, and shows clearly that surface roughness and etching duration affects SEY. Is there is a correlation with porosity? Figure 9 is difficult to follow, with limited discussion. I could not find Equation (3) and there are too many significant figures in the values of a, b, c and R squared (R2?). This section is not as well-structured or as clear as the other sections.
The Conclusions need more clarity, which should follow with better structure and clarity in the Discussion.
Reviewer 2 Report
Report on the manuscript entitled : « Effect of the surface morphology of porous coating on secondary electron yield of metal surface »
Min Peng, Shu Lin , Chuxian Zhang, Haifeng Liang, Chunliang Liu, Meng Cao1 , Wenbo Hu, Yonggui Zhai and Yongdong Li
The manuscript presents the influence of a 3-step surface treatment process for aluminum alloys, with Ni/Ag coating followed by etching and gold coating, allowing the secondary emission efficiency (SEY) to be reduced depending on the etching temperature and durations used. The effect of gold coating (to avoid surface oxidation particularly ) on SEY is also studied. An exponential fitting model giving evolution of SEYmax versus roughness parameter Ra is deduced from these measurements and allow further extrapolations to be made.
Although the relationship between surface roughness and SEY is well known, ways to reduce this emission, which can disrupt the operation and/or decrease the efficiency of aerospace devices (Hall thrusters for instance) are still under study. In this context, the contribution of the authors in this article is of great interest since they attempt to determine the conditions of treatment of the surface of irradiated materials to strongly reduce the secondary electronic emission and thus avoid the resulting deleterious electrons avalanche effects.
The content of the article remains in the technical field. The goal is to determine and ultimately predict the optimized surface treatment parameters of the coated sample to obtain a roughness likely to reduce the SEY.
The paper is well presented with concise introduction. The sketches and images are quite clear and understandable even though fig. 5 needs to be completed (see remark about SEY on Material and methods below) and some scans have a deteriorated resolution (fig. 4, 6). Nevertheless, it requires some revisions. Some minor revisions in the text with some questions and remarks are mixed below:
1. The text should be read again carefully. Please correct remaining mistakes and clarify some long sentences to help the readers.
Introduction:
22. I recommend adding more recent references.
3. Lines 60 to 63 : Please make this sentence clearer to better highlight the interest and contribution of the work
Materials and methods
4. Line 65 : An explanation for the choice of aluminum alloy 6061 for these experiments would be welcome. It is not only a substrate for porous coating.
5. Although the SEY measuring device has been described elsewhere (ref. 28 Weng et al for SEEY of insulators generated by a pulsed electron beam) it is necessary to recall and explain a little more about the experimental device and the measurement method. As measurement of SEY is not trivial, it should be explained and detailed carefully. From fig. 5 the sample seems irradiated by electrons at normal incidence, isn’t it ? - Considering the charge conservation law, it seems that primary current Ip should be the sum of Is and It and not the difference. The diagram of figure 5 must be completed : It must show for instance the Secondary electrons detector’s polarization (in Volt), the associated symbols (Ip, It, Is) and the direction of the various measured currents even though the ground is already represented.
6. As distribution of roughness is not necessarily uniform on the porous coating, does the little area of the coating, scanned by the color laser scanning microscope, corresponds to the electron irradiated zone for SEY measurement or not ?
Discussion
7. Line 171 : ‘’the common characteristics for low SEY were found to be granular clusters with around 1.0μm of pc values as well as the large surface Ra values…’’ should be replaced by : “the common characteristics for low SEY were found to be granular clusters with pc values around 1.0 μm as well as large surface Ra values..”
8. line 179 : “…in Figure 7, where also shows the statistical data for different schemes marked by colorful blocks. ‘’ : replace ’’where’’ by ‘’which’’ to make it more understandable.
9. Line 195 : ’’From the perspective of practical engineering, we pay more attention to what process scheme can obtain this kind of morphology conducive to suppress SEY.’’ Suppress « what » to clear signification of this sentence.
10. Line 208 : ‘’ This provides a basis of further probe into parameter schemes for surface morphology in pursuit of low SEY, and then Figure 9 presents a correspondence among etching parameter schemes, Ra values of resultant morphology and corresponding SEY suppression percentage when gold coating is 100 nm » : Please divide this paragraph in 2 sentences : ‘This provides a basis of further probe into parameter schemes for surface morphology giving low SEY. Figure 9 presents some correspondence between etching parameter schemes, Ra values of resultant morphology and corresponding SEY suppression percentage for a 100 nm thick gold coating.
11. Line 212 : ‘’ It is obvious that the region with strong suppression effect corresponds to large Ra values, and this region is concentrated in the center due to the mutual compensation of the two parameters. ‘’ Could the authors give precisions on this compensation ?
12. Emitted secondary electrons comes from gold or from nickel when comparison is made (with or without gold coating). At very low primary energy for the incident electrons beam bombardment, SEY depends on work function of atoms. This work function is higher for gold. Could the decrease of SEY, observed with gold coating for low etching duration and temperatures less than 25°C in fig. 8, be also attributed to this work function difference between gold and nickel ?
13. On fig.9, it is not mentioned in the text, but surface roughness parameter Ra is indeed deduced from the model of fig.8 ?
14. As electrons do not necessarily arrive at normal incidence on a material, did the authors measure the SEY for angles of incidence less than 90°?
